# Efficient Reasoning with Hidden Thinking

Xuan Shen [1]  Yizhou Wang [2]  Yufa Zhou [3]  Xiangxi Shi [4]  Pu Zhao [5]  Yanzhi Wang [5]  Jiuxiang Gu [2]

## Abstract

Chain-of-Thought (CoT) reasoning has become a powerful framework for improving complex problem-solving capabilities in Multimodal Large Language Models (MLLMs). However, the verbose nature of textual reasoning introduces significant inefficiencies. In this work, we propose **Heima** (as hidden llama), an effective CoT compression framework that condenses lengthy CoTs into a small set of abstract thinking tokens, preserving essential reasoning while removing redundancy. We then conduct a theoretical analysis from an information-theoretic perspective, quantifying the information gap induced by compression, showing that reasoning capability is preserved when non-trivial mutual information is retained. To further explore and quantify this information gap, we design the adaptive interpreter that maps thinking tokens back to variable-length textual sequences, thereby reconstructing the reasoning process. Experiments across diverse reasoning benchmarks demonstrate that Heima improves reasoning efficiency, while maintaining or even achieving better zero-shot accuracy. Moreover, the interpreter reconstructs coherent reasoning progresses from compressed thinking tokens, revealing that the information gap is minimal and validating the effectiveness of the proposed framework. This work paves the way for scalable latent reasoning models and advances our understanding of efficient reasoning processes in large models. Code: https://github.com/shawnricecake/Heima

---

[1]College of Computer Science and Technology, Zhejiang University, Hangzhou, Zhejiang, China [2]Adobe, San Jose, CA, USA [3]Department of Computer Science, Duke University, Durham, NC, USA [4]Oregon State University, Corvallis, OR, USA [5]Department of Electrical and Computer Engineering, Northeastern University, Boston, MA, USA. Correspondence to: Pu Zhao <p.zhao@northeastern.edu>, Jiuxiang Gu <gu.jiuxiang@gmail.com>.

*Proceedings of the 43rd International Conference on Machine Learning*, Seoul, South Korea. PMLR 306, 2026. Copyright 2026 by the author(s).

## 1. Introduction

The recent rise in popularity of Multimodal Large Language Models (MLLMs) (Achiam et al., 2023; Bai et al., 2023; Liu et al., 2024c; Lai et al., 2024; Xu et al., 2024; Shen et al., 2023; Zhao et al., 2024a; Shen et al., 2025b;e; 2026b), which integrate vision techniques with traditional Large Language Models (LLMs), has spurred interest in leveraging Chain-of-Thought (CoT) (Wei et al., 2022) reasoning to enhance their capabilities for solving complex problems. It not only enhances interpretability but also enables more effective multi-step reasoning, equipping MLLMs to address tasks that demand intricate logical understanding and contextual coherence, especially when processing the inherent complexity of visual information. However, reasoning with CoT often requires generating a substantial amount of additional reasoning texts, particularly for complex problems, leading to expensive inference costs. Thus, efficiency has become a central theme in applying MLLMs, making it crucial to reduce the number of tokens generated during reasoning to enhance overall computational performance.

Recent works (Hao et al., 2024; Deng et al., 2024b) explore the latent reasoning methods. Approach (Hao et al., 2024) explores compressing CoTs for a small-scale model, GPT-2 (Radford & Wu, 2019), on individual reasoning tasks in the text-only setting. However, this leaves a significant gap in extending CoT compression to large-scale MLLMs that must handle general reasoning tasks with multimodal inputs. Other works (Pi et al., 2023; Yan et al., 2024; Deng et al., 2024a) investigate latent reasoning in MLLMs by employing visual decoders for segmentation, detection, and recognition, aiming to decode latent information in MLLM generated tokens. This shows the feasibility of latent-space reasoning and motivates deeper investigation into its capabilities.

In this work, we propose Heima, the first CoT compression framework for MLLMs to achieve efficient reasoning. Instead of relying on verbose CoTs, Heima performs reasoning in the latent space, compressing CoTs into compact thinking tokens. First, we design Heima as the CoT compressor of our framework. Specifically, we train the reasoning MLLM to distill each CoT stage into a single thinking token, <CoT>, utilizing step-by-step distillation for more effective mapping. Then, we provide theoretical analysis from the information-theoretic perspective to quantify the

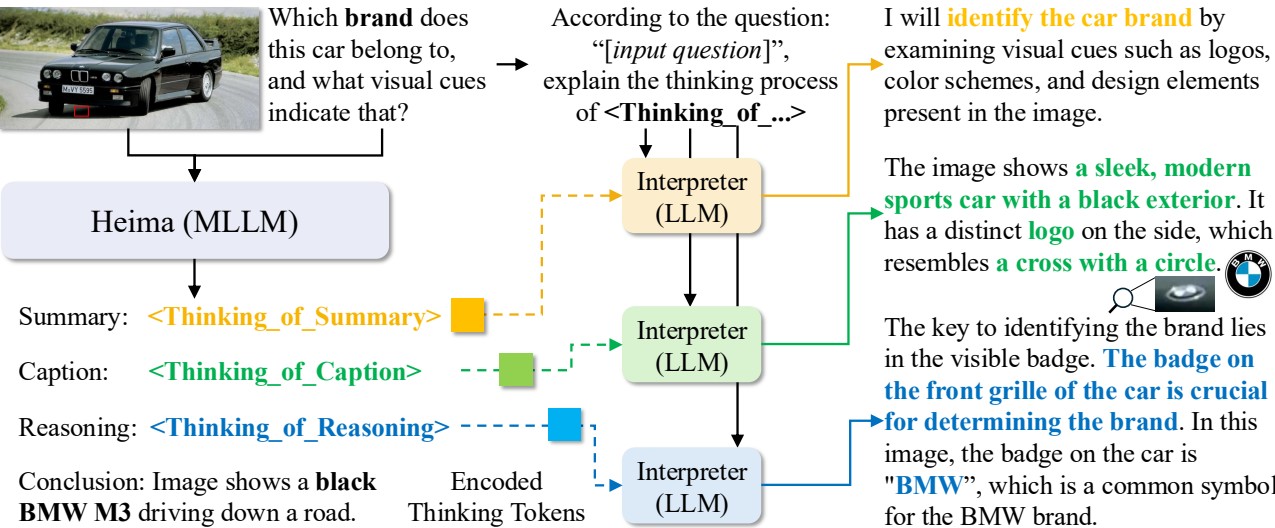

Figure 1. Heima (MLLM) performs reasoning with thinking tokens based on image and question. Compressed thinking tokens and question are then fed to interpreters (LLMs) for CoT reconstruction. In reconstructed reasoning progress, the caption interpreter successfully retrieves image information, describing it as *"The image shows a sleek, modern sports car with a black exterior"* and identifying a distinct feature of logo as *"a cross with a circle"*. Also, the reasoning interpreter accurately deduces the logo as BMW based on this distinctive symbol. **This verifies that interpreters effectively reconstruct visual features from pure textual inputs.**

information gap between textual CoTs and thinking tokens, showing that reasoning effectiveness is preserved when non-trivial mutual information is retained. To further assess and quantify this gap, we develop corresponding interpreters fine-tuned from LLMs that adaptively decode thinking tokens into textual sequences of varying length. By leveraging explanatory prompts, the interpreters effectively reconstruct reasoning progress from compressed representations.

The whole framework is shown in Figure 1. We accelerate the reasoning using Heima, which performs reasoning in latent space by generating a significantly reduced number of thinking tokens instead of verbose textual CoTs, thereby producing answers more efficiently. The last hidden states of these thinking tokens are further delivered to interpreters for the reconstruction of the textual reasoning sentences. Experimental results demonstrate that our approach significantly enhances reasoning efficiency by generating far fewer tokens while achieving comparable or even superior performance on a list of zero-shot reasoning benchmarks. Furthermore, the results demonstrate that the reasoning progress reconstructed by interpreters, even without visual information, closely align with the original CoTs texts generated based on multimodal inputs. These findings establish that the information gap induced by compression is negligible and affirm the effectiveness as well as the interpretability of the thinking tokens produced by Heima. Our contributions are summarized below:

- **1.** We propose Heima, the first reasoning acceleration framework for MLLMs that conducts reasoning in latent space by generating compact thinking tokens rather than verbose textual CoTs.
- **2.** We develop an information-theoretic analysis of

CoT compression within Heima that quantifies the compression-induced information gap and proves that reasoning capability is preserved under retention of non-trivial mutual information.

- **3.** We design interpreters with pure LLMs to reconstruct textual reasoning with thinking tokens, which serves to analyze the information gap induced by compression relative to the original CoTs.
- **4.** Experiments show that Heima attains substantial improvements in reasoning efficiency without sacrificing performance relative to textual CoTs. In addition, reconstructions with interpreters indicate that the information gap induced by compression is negligible, providing empirical validation of the theoretical guarantees underlying the Heima framework.

## 2. Related Work

### 2.1. Chain-of-Thought Reasoning

With the theoretically validated effectiveness in recent works (Merrill & Sabharwal, 2023; Feng et al., 2024; Zhou et al., 2026; Zhan et al., 2024b; Wang et al., 2025), CoTs have gained increasing popularity and are widely adopted as an enhancement method for generating intermediate reasoning processes before arriving at the final answer. The works (Wei et al., 2022; Khot et al., 2022; Zhou et al., 2022) focus on designing the effective prompts which decompose the question into a group of reasoning steps for LLMs. The works (Yue et al., 2023; Yu et al., 2023; Wang et al., 2023; Shao et al., 2024; Liu et al., 2024a) adopt additional fine-tuning to guide the model to generate reasoning chains. Meanwhile, to further enhance reasoning performance and

make model reasoning more human-like, recent works (Xie et al., 2024; Gandhi et al., 2024; Su et al., 2024) have integrated additional search algorithms to improve the relevance of CoT generation and the input content. However, almost all of these works enhance reasoning performance by generating additional textual tokens, which incurs significant computational costs for large generative models with billions of parameters. This motivates us to explore token reduction methods during reasoning process.

## 2.2. Textual Efficient Reasoning

Recent works (Ning et al., 2023; Kou et al., 2024; Zhang et al., 2023; Zhan et al., 2024a; Kong et al., 2025; Li et al., 2024; Shen et al., 2024a;b;c; 2025d; Zhao et al., 2025b; Lin et al., 2025; Shen et al., 2025c; 2026a; 2025a) aim to accelerate the reasoning process by employing parallel generation through templates or Jacobi decoding, which introduces additional overhead during model inference. Meanwhile, the works (Ge et al., 2024; Chevalier et al., 2023; Qin et al., 2023; Liu et al., 2023; Mi et al., 2026; Munkhdalai et al., 2024; Guan et al., 2021; Zhan et al., 2024c; Liu et al., 2025b;a;c; Nguyen et al., 2025; Zhao et al., 2025c;a; 2024b) adopt contextual compression methods to achieve the efficient generation of the next following contexts based on the previous contexts. On the other hand, the work (Cheng & Van Durme, 2024) compresses the CoT into a short sequence of continuous embeddings for the acceleration of reasoning process. However, its results on the math dataset reveal a significant degradation in accuracy, indicating the limitations and ineffectiveness of this approach. Thus, there is still a gap in developing efficient reasoning techniques for large models that maintain the advantages of reasoning with CoTs, which motivates us to explore better compression methods with CoTs for higher accuracy and efficiency.

## 2.3. Reasoning in Latent Space

Works (Hao et al., 2024; Zhao et al., 2026; Yang et al., 2024; Biran et al., 2024; Cheng & Van Durme, 2024) adopt latent reasoning for LLMs. For example, the work (Hao et al., 2024) addresses the compression of small models (GPT-2) on math datasets with CoTs. However, its effectiveness remains unverified as the evaluation is limited to math datasets and small model size. Some works (Liu et al., 2024c; Lai et al., 2024) include visual information into latent space for MLLMs to enhance the textual reasoning. The work (Lai et al., 2024) employs fine-tuning for both LLMs and segmentation decoders, demonstrating the potential to decode visual information in tokens generated by LLMs. Subsequent works (Pi et al., 2023; Deng et al., 2024a; Yan et al., 2024) continue to investigate using MLLMs to generate tokens for visual downstream tasks, rather than exploring the construction of internal feature representations within MLLMs for higher-level feature embedding. The absence

of feature construction in MLLMs motivates us to explore the development of latent representations for these models.

## 3. Methodology

We mainly present Heima framework in this section. First, we outline the training setup of Heima, covering dataset construction and the distillation strategy. Then, we provide theoretical analysis of the information gap between textual CoTs and compressed thinking tokens from an information-theoretic perspective. Next, we introduce the interpreter, which maps thinking tokens back to CoT-like trajectories, to further explore the information gap introduced by compression. Finally, we demonstrate how this approach enables efficient compressed reasoning at inference time.

### 3.1. Heima Framework

We introduce Heima to compress verbose textual CoTs into compact thinking tokens through further distillation, enabling faster on-the-fly CoT reasoning.

**Dataset Preparation.** Given $N$ total samples, original CoT training dataset $D$ is defined as follows,

$$D := \big\{(X, \text{CoTs}, Y)\big\}, \ |D| = N, \tag{1}$$

where $X$ denotes the visual and query inputs, $\text{CoTs} := \{\text{CoT}_{(k)}\}_{k=1}^{K_i}$ denotes the sequence of $K_i$ textual CoT stages in $i$-th sample, and $Y$ denotes the final answers.

**Thinking Token Dataset.** We define the reasoning MLLM as $\mathcal{H}$, which is fine-tuned on the updated dataset to incorporate CoTs. Specifically, we update the training set $D$ by replacing each $\text{CoTs}$ with thinking tokens—denoted as $\texttt{<CoTs>} := \big\{\texttt{<CoT>}_{(k)}\big\}_{k=1}^{K_i}$. For each thinking token $\texttt{<CoT>}_{(k)}$ at the $k$-th stage, it is defined with *a unique special token and added to the vocabulary* to enable explicit textual visualization of the reasoning process. For example, in Figure 1, we define a new token $\texttt{<Thinking\_of\_Summary>}$ in vocabulary as the thinking token for the summary stage. Note that different samples with varying values of $i$ share the same thinking token $\texttt{<CoT>}_{(k)}$ at the same stage $k$. The updated Heima dataset $D_H$ is then defined as follows,

$$D_H := \big\{(X, \texttt{<CoTs>}, Y)\big\}, \ |D_H| = N. \tag{2}$$

**Distillation Objective.** To perform the reasoning with thinking tokens in latent space, we further perform fine-tuning using CoT distillation. The distillation objective is defined as follows,

$$\mathcal{L}(\theta) = -\mathop{\mathbb{E}}_{(X,Y,\texttt{<CoTs>}) \sim D_H} \log P_\theta(\texttt{<CoTs>}, Y \mid X). \tag{3}$$

Through this distillation, we fine-tune model to directly predict thinking tokens $\texttt{<CoTs>}$ instead of verbose CoTs.

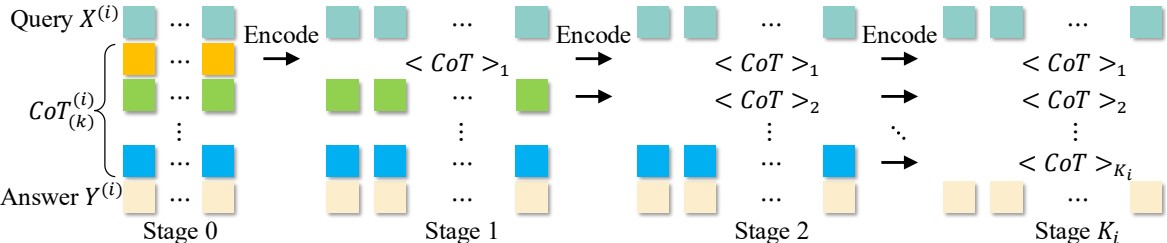

*Figure 2.* Visualization of progressive distillation. Each row shows the training data in that stage.

**Progressive Distillation.** To facilitate a seamless transition from textual reasoning to hidden thinking while maintaining model performance, we further adopt the progressive distillation strategy as shown in Figure 2. Specifically, we do not distill all CoT stages into thinking tokens at once. Instead, we *progressively* distill each stage into a thinking token, one by one. Formally, our progressive distillation has $M := \max\{K_i\}_{i=1}^N + 1$ stages. After we finish the $s$-th stage distillation, we move on to its next $(s+1)$-th stage distillation, until reaching the final stage. In the $s$-th stage ($s \in \mathbb{N}, 0 \le s < M$), we prepare the pregressive training data $|D_P| \le NM$ using

$$D_P := \left\{ \left( X, \{<\texttt{CoT}>_{(k)}\}_{k=1}^s, \{\text{CoT}_{(k)}\}_{k=s+1}^{K_i}, Y \right) \right\}. \quad (4)$$

For $s = 0$, the set $\{<\texttt{CoT}>_{(k)}\}_{k=1}^s$ is empty with no thinking tokens in training. For $s > 0$, we finetune Heima $\mathcal{H}_\theta(\cdot)$ as follows,

$$\max_\theta \mathbb{E} \log P_\theta \left( \{<\texttt{CoT}>_{(k)}\}_{k=1}^s, \{\text{CoT}_{(k)}\}_{k=s+1}^{K_i}, Y \mid X \right). \quad (5)$$

In the above expression, during the $s$-th stage distillation, the first $s$ CoT stages are distilled with thinking tokens, while the rest CoT stages are untouched and trained with textual reasoning tokens. As the value of $s$ increases, more CoT stages are distilled into thinking tokens as shown in Figure 2.

This approach allows the model to gradually internalize the reasoning processes with multiple CoTs and integrate them into thinking tokens. After the final progressive stage, we introduce an additional *recovering stage*, where the model is further optimized using only thinking tokens as in Equation (3). This extra recovering stage optimizes the transitions and interactions between the thinking tokens across different stages, ensuring a cohesive alignment of information learned throughout the distillation process. Additionally, it consolidates the overall learning process, enhancing the model's ability to effectively utilize the distilled reasoning patterns for improved performance and robustness in downstream reasoning tasks.

**Efficient Reasoning.** To accelerate reasoning with Heima framework, we deploy Heima solely as the service reasoning MLLM, benefiting from lower memory usage and faster generation. Heima generates the response corresponding to the given input as follows,

$$P_\theta\left(<\texttt{CoTs}>, Y \mid X\right) = \prod_{t=1}^{T+K_i} P_\theta\left(w_t \mid w_{<t}, X\right), \quad (6)$$

where the $(w_1, w_2, \ldots, w_{K_i}) = <\texttt{CoTs}>$ denotes the generated thinking tokens, and the $Y = (w_{K_i+1}, w_{K_i+2}, \ldots, w_{K_i+T})$ denotes the tokens corresponding to the final answer. We obtain the final answer by Heima using only $K_i$ intermediate tokens, significantly reduced from the original $\sum_{k=1}^{K_i} |\text{CoT}_{(k)}|$ tokens, enabling faster generation and avoiding complex & verbose textual CoTs.

### 3.2. Information-Theoretic Compression

Distillation (Equation (3)) represents a form of compression (Delétang et al., 2024; Huang et al., 2024), since such loss induces a code when paired with an ideal entropy coder (MacKay, 2003). Extending this view, we provide an analysis of information variance across the pre- and post-distillation stages.

**Notations.** For a random variable $X \sim P_X$, entropy is $H(X) = -\mathbb{E}_{x \sim P_X}[\log P_X(x)]$. For joint variables $(X, Y)$ with distribution $P_{X,Y}$, conditional entropy is $H(Y \mid X) = \mathbb{E}_{x \sim P_X}[H(P_{Y|X=x})]$. Mutual information is $I(X; Y) = H(X) - H(X \mid Y) = \text{KL}(P_{X,Y} \parallel P_X P_Y)$, and conditional mutual information is $I(X; Y \mid Z) = H(X \mid Z) - H(X \mid Y, Z)$.

Given input $X$ for inference, Heima compresses textual CoTs into compact representations $<\texttt{CoTs}> = f(X, \text{CoTs})$ as thinking tokens with $|<\texttt{CoTs}>| \ll |\text{CoTs}|$. The central question is whether $<\texttt{CoTs}>$ retains sufficient task-relevant information about the output $Y$. By the data-processing inequality (MacKay, 2003) as follows,

$$0 < I(Y; <\texttt{CoTs}> \mid X) \le I(Y; \text{CoTs} \mid X),$$

which shows that $<\texttt{CoTs}>$ carry no more information than CoTs, while still preserving non-trivial information for reasoning. This motivates Heima as a compressor of CoTs, a property we formalize in Theorem 3.1.

**Theorem 3.1** (Information preserved under CoT compression). *Let $X$ be the input,* CoTs *chain-of-thoughts,* $\texttt{<CoTs>} = f(X, \text{CoTs})$ *thinking tokens, and $Y$ the task output. Since $\texttt{<CoTs>} = f(X, \text{CoTs})$, we have the Markov chain $Y - (X, \text{CoTs}) - \texttt{<CoTs>}$. Then*

$$H(Y \mid X, \text{CoTs}) \leq H(Y \mid X, \texttt{<CoTs>}) \leq H(Y \mid X),$$

*equivalently,*

$$0 \leq I(Y; \texttt{<CoTs>} \mid X) \leq I(Y; \text{CoTs} \mid X),$$
$$and \quad I(Y; \text{CoTs} \mid X) - I(Y; \texttt{<CoTs>} \mid X)$$
$$= I(Y; \text{CoTs} \mid X, \texttt{<CoTs>}) \geq 0.$$

*Thus, compression into $\texttt{<CoTs>}$ cannot increase the information about $Y$ relative to* CoTs. *It preserves strictly positive task-relevant information if $I(Y; \texttt{<CoTs>} \mid X) > 0$, and loses no information compared to* CoTs *if and only if $I(Y; \text{CoTs} \mid X, \texttt{<CoTs>}) = 0$ (i.e., $\texttt{<CoTs>}$ is sufficient for* CoTs *with respect to $Y$).*

*Proof.* Because $\texttt{<CoTs>} = f(X, \text{CoTs})$, $Y - (X, \text{CoTs}) - \texttt{<CoTs>}$ holds, and by the data processing inequality (MacKay, 2003) $I(Y; \texttt{<CoTs>} \mid X) \leq I(Y; \text{CoTs} \mid X)$. Using the identity $I(Y; W \mid X) = H(Y \mid X) - H(Y \mid X, W)$ for $W \in \{\texttt{<CoTs>}, \text{CoTs}\}$ gives $H(Y \mid X, \text{CoTs}) \leq H(Y \mid X, \texttt{<CoTs>}) \leq H(Y \mid X)$. Finally, the chain rule with $I(Y; \texttt{<CoTs>} \mid X, \text{CoTs}) = 0$ yields $I(Y; \text{CoTs} \mid X) = I(Y; \texttt{<CoTs>} \mid X) + I(Y; \text{CoTs} \mid X, \texttt{<CoTs>})$, proving the nonnegative gap and sufficiency condition. $\square$

*Remark* 3.2. This result formalizes the role of CoT compression: the compressed $\texttt{<CoTs>}$ can never be more informative about the target $Y$ than the full chain CoTs, but as long as $I(Y; \texttt{<CoTs>} \mid X) > 0$, it still retains strictly useful information beyond $X$ alone. The gap $I(Y; \text{CoTs} \mid X, \texttt{<CoTs>})$ quantifies the reasoning details lost in compression, while the inequality ensures that compressed reasoning remains effective as long as $\texttt{<CoTs>}$ captures nontrivial task-relevant information.

Theorem 3.1 and Remark 3.2 indicate that the gap $I(Y; \text{CoTs} \mid X, \texttt{<CoTs>})$ determines the amount of information loss during the compression from verbose textual CoTs to compact thinking tokens. This observation highlights that the effectiveness of reasoning in latent space fundamentally depends on how much of the task-relevant information is preserved after compression.

### 3.3. Interpreter Design

As explained in the above section, it is essential to analyze and quantify this information gap in order to assess whether the compressed representations remain sufficient for reasoning. To this end, we further design the corresponding interpreters that reconstruct thinking tokens back into textual reasoning traces, thereby providing an empirical proxy to evaluate the magnitude of this gap and verify the practical validity of reasoning in latent space. Meanwhile, this can verify if the model is genuinely learning reasoning in latent space rather than merely fitting the data, showing the effectiveness of compressed representations encapsulated within thinking tokens. Specifically, for the design of interpreter, we adopt the standard next-token prediction objective in pure LLMs for the reconstruction of variable-length (i.e., adaptive) textual sequences based on thinking tokens.

**Adaptive Interpretation.** After full distillation in Heima, all CoT stages are distilled into thinking tokens. Thinking token for each reasoning stage requires one corresponding interpreter. Thus, to interpret all of the thinking tokens, we train the corresponding interpreters separately. In detail, we adopt a pretrained LLM as the initialization of interpreter $\mathcal{I}_{\theta_k}(\cdot)$ for the $k$-th CoT stage corresponding to the $k$-th thinking token $\texttt{<CoT>}_{(k)}$, where $\theta_k$ denotes its parameters. We do not consider the case of $k = 0$ without thinking tokens. Its training set $D_I$ of the $k$-th stage is designed as follows, for $|D_I| = N$, let

$$D_I := \left\{ (X_e, X_q, \texttt{<CoT>}_{(k)}, H_{\texttt{<CoT>}_{(k)}}, \text{CoT}_{(k)}) \right\}, \quad (7)$$

where $X_e$ denotes the explanatory prompts to guide the model for the interpretation of thinking tokens. $X_q$ denotes the pure textual question. $H_{\texttt{<CoT>}_{(k)}}$ denotes the hidden representation (last hidden states) of the thinking token $\texttt{<CoT>}_{(k)}$ generated by Heima. $\text{CoT}_{(k)}$ denotes the original textual CoT at $k$-th stage. Note that, here we only use pure-text LLMs as the interpreters, meaning they cannot read images. The dataset $D_I$ for training interpreters only has text questions without visual images inputs.

During the training of interpreter, the frozen Heima is used to generate the thinking tokens. Importantly, we do not feed the token symbol $\texttt{<CoT>}_{(k)}$ directly as input. Instead, we replace it with the corresponding last hidden state $H_{\texttt{<CoT>}_{(k)}}$ from Heima, since the reasoning information is encapsulated in the hidden representation (i.e., last hidden state) rather than the textual symbol. This substitution occurs after the word embedding stage, as illustrated in Figure 3. Interpreter is fine-tuned with the next-token prediction loss as follows,

$$\max_{\theta_k} \mathbb{E} \log P_{\theta_k} \left( \text{CoT}_{(k)} \mid X_e, X_q, H_{\texttt{<CoT>}_{(k)}} \right). \quad (8)$$

**Explanatory Prompts.** The single hidden representation $H_{\texttt{<CoT>}_{(k)}}$ alone is insufficient to guide the interpreter $\mathcal{I}_{\theta_k}(\cdot)$ toward reconstructing the original reasoning texts, as language models generally rely on textual instructions to scaffold the generation process. Thus, we provide the explanatory prompt for interpreter to enhance usability. We use the prompt as follows,

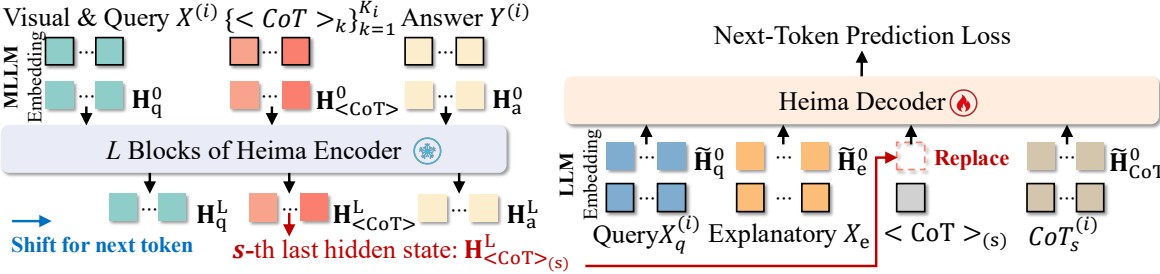

*Figure 3.* Training progress for the interpreter. Thinking token is caught from the $s$-th last hidden state of Heima and replaces the embedding of special token $\texttt{<CoT>}_{(s)}$.

> *"According to question: $X_q$, can you explain the thinking progress $\texttt{<CoT>}_{(k)}$?"*

This kind of prompt ensures that the output reasoning process remains (i) aligned with the original query $X_q$ and (ii) consistent with the hidden representations contained in thinking tokens.

The interpreter is employed to investigate the information gap highlighted in Theorem 3.1 and Remark 3.2 by comparing the reconstructed reasoning sentences with the original textual CoTs. When the reconstructed reasoning closely aligns in semantics with the original CoTs, the compression-induced information gap is regarded as minimal, thereby confirming that reasoning with thinking tokens preserves the essential reasoning capability.

## 4. Experimental Results

### 4.1. Experiment Setup

**Dataset.** We utilize the $\texttt{LLaVA-CoT-100k}$ (Xu et al., 2024) dataset, a reasoning dataset for MLLMs that integrates samples from several widely used VQA datasets. It comprises 100k image-QA pairs with three stages of CoT reasoning: summary, caption, and reasoning.

**Model Training.** We adopt the LLaVA-CoT (Xu et al., 2024) pretrained model based on Llama-3.2-11B-Vision-Instruct (Meta, 2024b) as the initialization of Heima. The Llama-3.1-8B-Instruct (Meta, 2024a) is employed as the initialization of interpreter. We use torchtune (Meta, 2024c) as the model training framework with LoRA (Hu et al., 2021) for both Heima and the corresponding interpreter. During the progressive distillation, we freeze the image encoder component and fine-tune both the decoder module and fusion components of the LLaVA-CoT model. This distillation includes the entire attention and MLP modules across all layers, as well as the output projection layer, using a rank of 16, and an alpha of 32. For the training of interpreter, we apply the same LoRA setting. Detailed hyperparameters are included in Appendix A. The training is conducted on $8\times$ H100 GPUs. Besides, to further verify our generalization for different model architectures, the LLaVA-Next-Vicuna-

7B (Liu et al., 2024b) (as Heima) and Vicuna-7B (Zheng et al., 2023) (as interpreter) are adopted. We train LLaVA-Next-Vicuna-7B on $\texttt{LLaVA-CoT-100k}$ through LoRA to capture reasoning capability, and then perform our method with thinking tokens with this model family.

**Evaluation.** We adopt multiple challenging zero-shot benchmarks to verify the effectiveness of our proposed method, including MMStar (Chen et al., 2024), MMBench V1.1 (Liu et al., 2025d), MMVet (Yu et al., 2024), Math-Vista (Lu et al., 2024), AI2D (Hiippala et al., 2021), and HallusionBench (Guan et al., 2024). MMStar, MMBench, and MMVet evaluate general visual question-answering capabilities, while MathVista and AI2D assess mathematical and scientific reasoning. HallusionBench, in contrast, targets language hallucinations and visual illusions. We use the VLMEvalKit (Duan et al., 2024) as the evaluation pipeline to ensure a fair comparison. We reproduce the evaluation results of LLaVA-CoT to get the number of generated tokens. GPT-4o (Achiam et al., 2023) is adopted for evaluation on the MMVet and MathVista datasets, while exact match evaluation is applied to other datasets using VLMEvalKit. For Heima, we split the $\texttt{LLaVA-CoT-100k}$ dataset for train and test separately. We evaluate fine-tuned interpreters on test set which contain 4300 samples with metrics including BLEU-4 (Papineni et al., 2002), METEOR (Banerjee & Lavie, 2005), ROUGE (Lin, 2004), BERTScore (Zhang et al., 2019), and similarity analysis from GPT-4o.

### 4.2. Main Results

We first provide main results for Heima in Table 1. We compare our method with original Llama3.2-11B-Vision-Instruct and the LLaVA-CoT on 6 datasets for zero-shot evaluation. Heima outperforms Llama3.2-11B-Vision-Instruct model with large improvements in average accuracy, while using fewer tokens, particularly on benchmarks such as MMBench, AI2D, and Hallusion with much higher accuracy and fewer tokens. Compared with baseline LLaVA-CoT, Heima retains most of the model's performance while using as little as 6% of the tokens on certain datasets. Notably, on MM-Bench, Heima achieves better accuracy than the baseline LLaVA-CoT. Furthermore, to demonstrate the effectiveness

*Table 1.* Main results compared to Llama3.2-11B-Vision-Instruct and LLaVA-CoT with both accuracy and number of generated tokens on 6 different multimodal reasoning benchmarks.

| Dataset | MMSar | MMBench | MMVet | MathVista | AI2D | Hallusion | Avg. |
|---|---|---|---|---|---|---|---|
| Model | Acc. (# Token) | Acc. (# Token) | Acc. (# Token) | Acc. (# Token) | Acc. (# Token) | Acc. (# Token) | Acc. |
| Llama3.2 -11B Vision | 48.1 (140.0) | 58.2 (64.7) | 50.2 (106.0) | 50.3 (240.1) | 68.5 (74.9) | 37.2 (91.4) | 52.1 |
| LLaVA-CoT | 54.0 (181.0) | 70.7 (154.8) | 49.8 (227.2) | 50.9 (216.3) | 77.6 (178.5) | 63.8 (177.9) | 61.1 |
| Heima w/o progressive | 49.7 (13.1) | 72.5 (13.3) | 39.0 (71.7) | 39.3 (13.6) | 75.9 (12.6) | 61.3 (15.6) | 56.3 |
| Heima w/o recover | 49.8 (13.0) | 71.6 (13.2) | 42.8 (79.6) | 39.8 (14.0) | 77.3 (12.7) | 58.5 (17.5) | 56.6 |
| **Heima** | 49.9 (12.8) | 72.8 (12.9) | 43.3 (75.8) | 43.6 (13.8) | 77.5 (12.7) | 60.6 (16.9) | **58.0** |

*Table 2.* Results with LLaVA model family. LoRA is used to train the model for CoT improvements.

| Datasets | MMSar | MMBench | MMVet | MathVista | AI2D | Hallusion | Avg. |
|---|---|---|---|---|---|---|---|
| Model | Acc. (# Token) | Acc. (# Token) | Acc. (# Token) | Acc. (# Token) | Acc. (# Token) | Acc. (# Token) | Acc. |
| LLaVA-Next -Vicuna-7B | 37.7 (2.5) | 65.6 (2.0) | 33.4 (143.9) | 30.2 (93.7) | 67.0 (2.0) | 32.3 (69.3) | 44.4 |
| LLaVA-Next -Vicuna-7B (CoT) | 46.5 (175.9) | 71.5 (155.3) | 47.5 (230.5) | 41.8 (190.6) | 77.3 (165.7) | 45.1 (149.8) | 55.0 |
| **Heima** | 44.6 (12.8) | 73.5 (12.5) | 43.4 (68.9) | 40.6 (12.7) | 77.1 (12.6) | 43.3 (15.7) | 53.8 |

of progressive distillation, we show the accuracy results using one-shot distillation to distill all CoT stages in one shot. Results achieved with non-progressive distillation indicate worse performance, confirming the effectiveness of the progressive distillation in our framework. Additionally, the accuracy results without the recovering stage highlight its necessity, as they demonstrate a noticeable decline in performance compared to that with the recovering stage after completing the distillation of all CoT stages. We further present detailed accuracy results for various reasoning tasks of MMStar in Table A4 of Appendix B. Heima outperforms Llama3.2-11B on both instance reasoning (IR) and logical reasoning (LR) tasks while using less than 10% of tokens, and it preserves the majority of its reasoning capabilities for mathematical problems through progressive distillation.

Meanwhile, we provide additional results with LLaVA-Next-Vicuna-7B in Table 2 to verify the generalization of our framework. Our method achieves better performance than non-CoT model (i.e., the original LLaVA-Next-Vicuna-7B). Compared with the LoRA fine-tuned CoT model, our method achieves comparable accuracy with significantly fewer generated tokens (as little as 6%). The consistent performance on different models architectures demonstrates the effectiveness, efficiency, and generalization of our method.

### 4.3. Interpretability Analysis

To quantify the information gap by the compression as illustrated in Section 3.2, we evaluate the similarity between interpreter-reconstructed reasoning sentences and the ground-truth textual CoTs. We provide results of 4 evaluation metrics in the left side of Figure 4 with details in Table A5 of Appendix B. We observe that the reconstruction is most successful for summary stage, followed by caption stage, and then reasoning stage. In addition, we use GPT-4o to evaluate the similarity between reconstructed reasoning and original CoTs using a 5-point ranking scale (higher is better), and the results are shown in the right side of Figure 4. Prompts for GPT-4o are included in Appendix C. We average the rank of all samples in one stage to estimate the similarity score, and results verifies that all stages are effectively reconstructed.

Meanwhile, we provide evaluation results for interpreter with the LLaVA model family in Table A6 of Appendix B. Results are consistent with those in Figure 4 from Llama3 model family, further demonstrating the effectiveness and generalization of our method.

The results discussed above demonstrate that the information gap introduced by Heima between verbose textual CoTs

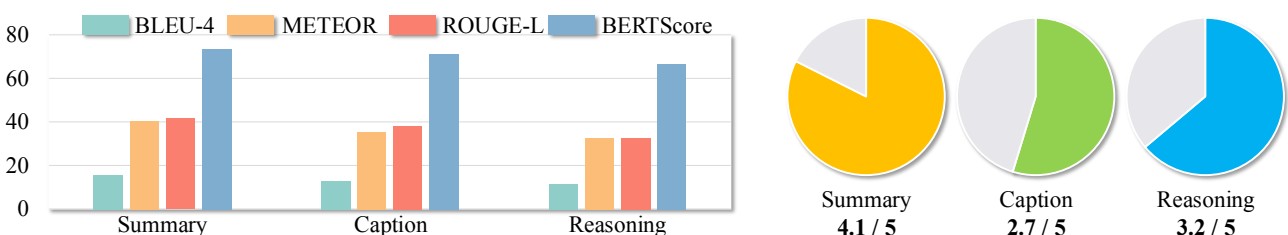

Figure 4. **Left**: results of BLEU-4, METEOR, GROUGE-L, and BERTScore for 3 interpreters. **Right**: results of evaluation by GPT-4o for evaluating the average similarity score (1-5) between the reconstructed reasoning processes from thinking tokens and the original textual CoTs.

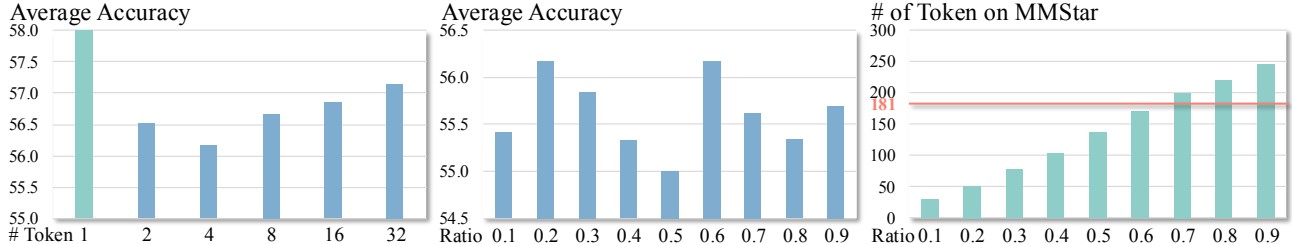

Figure 5. **Left**: ablation study of zero-shot performance on 6 datasets for different number of thinking tokens for each CoT. **Mid**: ablation study of average accuracy on 6 datasets for varying retention ratios of thinking tokens relative to original CoT. **Right**: ablation study for the number of generated tokens on MMStar with varying retention ratios of thinking tokens relative to the original CoT. Baseline (i.e., LLaVA-CoT) generates 181 tokens in average.

and compact thinking tokens is minimal, thereby validating the theoretical analysis in Section 3.2. More importantly, the reasoning capability of the compressed representation is well preserved, indicating that the retained non-trivial mutual information is sufficient for sustaining effective reasoning in latent space. In particular, the example in Figure 1 highlights that the interpreter successfully reconstructs textual reasoning progress that capture the key insights of visual information even when no visual input is provided. This finding confirms that thinking tokens preserve multimodal signals for reasoning, underscoring both the effectiveness of the compression with Heima and the robustness of reasoning in latent space.

### 4.4. Ablation Study

We provide results with different numbers of thinking tokens adopted for the distillation of one CoT stage in left side of Figure 5 with detailed results in Table A7 of Appendix B. Results show that the single thinking token for encoding CoT stage achieves the best performance.

We further ablate for adaptive distillation by using different retention ratio of the original CoT token length, and the corresponding results are in middle of Figure 5 with detailed results in Table A8 of Appendix B. Across retention ratios ranging from 10% to 90%, the accuracy exhibits irregular fluctuations without a discernible trend, reflecting the inherently unpredictable relationship between retention ratio and accuracy. Also, as shown in right side of Figure 5, the average number of generated token keeps increasing as

retention ratio becomes larger. When retention ratio reaches 70%, the average number of generated tokens exceeds that of baseline model, indicating the adaptive distillation is not effective for the compression of the reasoning progress.

In addition, we conduct the ablation study on the number of interpreters to examine whether it is necessary to adopt distinct interpreters for each CoT stage. The results are reported in Table A9 in Appendix B.5. The ablation results reveal that distinct interpreters are crucial for summary and caption reconstruction. We emphasize that interpreters are not integrated into the efficient reasoning procedure of Heima, but are instead utilized exclusively to interpret the reasoning in latent space.

## 5. Conclusion

We introduced Heima, a framework for accelerating reasoning in MLLMs through CoT compression. Heima distills each CoT into a compact thinking token and is supported by an information-theoretic analysis that quantifies the compression-induced information gap. To empirically examine this gap, we further design the interpreters guided by explanatory prompts to reconstruct reasoning progresses from thinking tokens. Extensive experiments demonstrate that Heima achieves comparable or even superior zero-shot accuracy with significantly fewer tokens, highlighting both its efficiency and robustness. Moreover, the successful reconstruction of reasoning processes confirms that the information gap is minimal and reasoning capability is preserved. Looking ahead, we will extend Heima to larger models.

## Impact Statement

This paper presents work whose goal is to advance the field of Machine Learning. There are many potential societal consequences of our work, none which we feel must be specifically highlighted here.

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

# Appendix

## A. Training Hyperparameters

We provide the hyperparameters for the progressive distillation, additional recovering, and adaptive interpretation training in Table A1, Table A2, and Table A3.

*Table A1.* Training hyperparameters for progressive distillation.

| Parameter | Value |
| --- | --- |
| Epoch | 1 |
| Batch Size | 6 |
| Gradient Accumulation | 1 |
| Optimizer | AdamW |
| Weight Decay | 0.01 |
| Learning Rate | 1e-4 |
| Learning Rate scheduler | cosine |
| Warmup | 100 |
| Clip Gradient Norm | 1 |
| Activation Checkpointing | TRUE |
| FSDP | TRUE |
| Bfloat16 | TRUE |

*Table A2.* Training hyperparameters for additional recovering after progressive distillation.

| Parameter | Value |
| --- | --- |
| Epoch | 1 |
| Batch Size | 8 |
| Gradient Accumulation | 1 |
| Optimizer | AdamW |
| Weight Decay | 0.01 |
| Learning Rate | 1e-5 |
| Learning Rate scheduler | cosine |
| Warmup | 100 |
| Clip Gradient Norm | 1 |
| Activation Checkpointing | TRUE |
| FSDP | TRUE |
| Bfloat16 | TRUE |

## B. Additional Results

### B.1. Detailed Results on MMStar

We present the detailed accuracy results on MMStar in Table A4, we identify that Heima outperforms Llama3.2-11B on both instance reasoning (IR) and logical reasoning (LR) tasks while using less than 10% of the tokens, and it preserves the majority of its reasoning capabilities for mathematical problems through progressive distillation.

### B.2. Detailed Evaluation of Interpreters

We provide the detailed evaluation results for the metrics in Table A5.

### B.3. Results with LLaVA Model Family

We provide the results of 3 interpreters trained based on Vicuna-7B in Table A6.

*Table A3.* Training hyperparameters for adaptive interpretation training.

| Parameter | Value |
|---|:---:|
| Epoch | 1 |
| Batch Size | 8 |
| Gradient Accumulation | 1 |
| Optimizer | AdamW |
| Weight Decay | 0.01 |
| Learning Rate | 5e-4 |
| Learning Rate scheduler | cosine |
| Warmup | 100 |
| Clip Gradient Norm | 1 |
| Activation Checkpointing | TRUE |
| FSDP | TRUE |
| Bfloat16 | TRUE |

*Table A4.* MMStar detailed results. CP denotes coarse perception, FP denotes fine-grained perception, IR denotes instance reasoning, LR denotes logical reasoning, S&T denotes Science&Technology. Overall accuracy is a weighted metric based on sample counts.

| Model | CP | FP | **IR** | **LR** | **Math** | S & T | Overall |
|---|:---:|:---:|:---:|:---:|:---:|:---:|:---:|
| Llama3.2 -11B Vision | 64.0 | 39.2 | 53.6 | 51.6 | 51.6 | 28.4 | 48.1 |
| LLaVA-CoT (reproduce) | 66.0 | 40.0 | 64.4 | 52.4 | 60.8 | 40.4 | 54.0 |
| Heima w/o progressive | 66.0 | 43.2 | 62.4 | 45.6 | 44.8 | 36.0 | 49.7 |
| Heima w/o recover | 64.8 | 44.0 | 57.2 | 51.6 | 44.0 | 37.2 | 49.8 |
| **Heima** | 62.0 | 43.2 | 58.8 | 52.8 | 48.0 | 34.8 | **49.9** |

## B.4. Detailed Ablation Study

We provide the detailed evaluation results of the ablation study for the different number of thinking tokens and different retention ratios in Table A7 and Table A8, separately.

## B.5. Ablation Study for Number of Decoders

We provide additional ablation study for using one LLM as the interpreter of 3 stages with BERTScore metric in Table A9. The evaluation results indicate that the single interpreter performs well on the reasoning stage but poorly on both the summary and caption stages, highlighting the necessity of employing separate interpreter for summary and caption stages.

*Table A5.* Detailed evaluation metrics for 3 interpreters trained based on LLama-3.1-8B-Instruct.

| Stage | Summary | Caption | Reasoning |
|---|---|---|---|
| BLEU | 15.9 | 12.8 | 11.2 |
| METEOR | 40.1 | 35.5 | 32.7 |
| ROUGE-L | 41.6 | 37.9 | 32.7 |
| BERTScore | 73.4 | 71.4 | 66.6 |

*Table A6.* Evaluation results for 3 interpreters trained based on Vicuna-7B.

| Stage | Summary | Caption | Reasoning |
|---|---|---|---|
| BLEU | 14.5 | 13.2 | 10.6 |
| METEOR | 39.5 | 31.5 | 31.5 |
| ROUGE-L | 42.1 | 34.1 | 30.6 |
| BERTScore | 69.6 | 65.7 | 64.9 |

*Table A7.* Detailed results for the ablation study of different number of thinking tokens.

| # Token | MMSar | MMBench | MMVet | MathVista | AI2D | Hallusion | Avg. Acc. |
|---|---|---|---|---|---|---|---|
| 1 | 49.9 | 72.8 | 43.3 | 43.6 | 77.5 | 60.6 | 58.0 |
| 2 | 50.3 | 71.4 | 41.4 | 43.1 | 75.6 | 57.3 | 56.5 |
| 4 | 49.9 | 71.0 | 42.2 | 39.3 | 75.4 | 59.3 | 56.2 |
| 8 | 51.1 | 70.4 | 41.0 | 40.9 | 76.7 | 59.9 | 56.7 |
| 16 | 49.5 | 72.0 | 40.9 | 40.9 | 76.2 | 61.6 | 56.9 |
| 32 | 50.2 | 71.1 | 42.9 | 41.6 | 75.2 | 61.8 | 57.1 |

*Table A8.* Detailed results for the ablation study of different retention ratios.

| Ratio | MMSar | MMBench | MMVet | MathVista | AI2D | Hallusion | Avg. Acc. |
|---|---|---|---|---|---|---|---|
| 0.1 | 49.1 | 69.7 | 37.2 | 41.3 | 75.9 | 59.1 | 55.4 |
| 0.2 | 49.7 | 71.5 | 39.4 | 41.2 | 75.3 | 60.0 | 56.2 |
| 0.3 | 48.1 | 71.9 | 40.6 | 39.9 | 75.3 | 59.4 | 55.8 |
| 0.4 | 47.9 | 70.3 | 38.6 | 39.2 | 76.3 | 59.7 | 55.3 |
| 0.5 | 47.2 | 70.1 | 40.5 | 39.5 | 75.2 | 57.6 | 55.0 |
| 0.6 | 48.4 | 70.9 | 42.0 | 38.8 | 76.6 | 60.5 | 56.2 |
| 0.7 | 48.7 | 69.8 | 41.1 | 39.0 | 75.4 | 59.7 | 55.6 |
| 0.8 | 49.9 | 69.3 | 40.9 | 37.2 | 75.3 | 59.4 | 55.3 |
| 0.9 | 49.2 | 70.5 | 40.1 | 38.4 | 75.7 | 60.1 | 55.7 |

*Table A9.* Ablation results on BERTScore metric for number of interpreters corresponding to 3 stages trained based on LLama-3.1-8B-Instruct.

| Metric | # of Decoders | Summary | Caption | Reasoning |
|---|---|---|---|---|
| BLEU | 1 | 9.5 | 6.8 | **11.3** |
| | 3 | **15.9** | **12.8** | 11.2 |
| METEOR | 1 | 32.7 | 25.4 | 32.5 |
| | 3 | **40.1** | **35.5** | **32.7** |
| ROUGE-L | 1 | 36.8 | 29.7 | 31.9 |
| | 3 | **41.6** | **37.9** | **32.7** |
| BERTScore | 1 | 67.8 | 60.6 | **67.3** |
| | 3 | **73.4** | **71.4** | 66.6 |

# C. Prompts for GPT-4o Evaluation

We provide the GPT-4o prompts in Prompt A1. In detail, we treat the evaluation as a ranking process to classify the performance of the reconstructed reasoning process into 5 ranks, from 1 to 5. Rank 1 represents a reconstructed reasoning process with little overlap with the ground-truth CoT and describing different themes, while Rank 5 represents a reconstruction that is well aligned with the ground truth. We remove special tokens from both sides and input them to GPT-4o to rank the similarity at each stage. We also include the corresponding image-question pairs in the prompt as additional references for more accurate contextual support.

---

**CoT Reconstruction Evaluation Agent (GPT-4o)**

**Input:** Image $\mathbf{I}$, question $\mathbf{Q}$, reconstructed CoT $\hat{\mathbf{CoT}}$, ground-truth CoT $\mathbf{CoT}$, and CoT stage type $\mathbf{T} \in$ {caption, summary, reasoning}.
**Output:** An integer similarity rank in $\{1, 2, 3, 4, 5\}$ measuring the alignment between $\hat{\mathbf{CoT}}$ and $\mathbf{CoT}$.

---

**Agent Role**
You are an evaluation agent responsible for assessing whether a generated analysis aligns with the ground truth, given the image–question pair. The analysis corresponds to one of the following stages:

- **Caption**: Description of the image content.

- **Summary**: Restatement or paraphrase of the question.

- **Reasoning**: Logical explanation linking the image and question to the answer.

**Evaluation Task**
You are given:

- Generated stage $\mathbf{T}$: $\hat{\mathbf{CoT}}$

- Ground truth: $\mathbf{CoT}$

- Image $\mathbf{I}$ and question $\mathbf{Q}$

Evaluate how closely the generated $\mathbf{T}$ matches the ground truth according to the image and question.
**Similarity Scale**

- **1 — Completely unrelated**: No thematic or factual overlap.

- **2 — Minimally related**: Only weak or tangential overlap.

- **3 — Partially related**: Core theme overlaps but with clear factual errors or missing key details.

- **4 — Closely related**: Main content aligns with minor discrepancies.

- **5 — Nearly identical**: Almost complete alignment with negligible differences.

**Response Format**
Return a JSON object: { "**T**" : Rank, "*reason*" : explanation } where `Rank` is the integer similarity score and `reason` briefly justifies the decision.

---

*Figure A1.* Instruction prompts for the Reconstruction Evaluation Agent using GPT-4o.

