# OpenReview forum: "Efficient Reasoning with Hidden Thinking"
_ICML.cc/2026/Conference — ICML 2026 regular_

### Official Review · Reviewer_n63E · 2026-03-03

**Soundness:** 3
**Presentation:** 2
**Significance:** 2
**Originality:** 2
**Overall Recommendation:** 4
**Confidence:** 3

**Summary:**

This paper proposes Heima, a multi-modal model applying efficient latent reasoning on visual tasks. Heima is trained on llava-cot and expresses comparable performance to MLLM baslines, yet requires much fewer CoT tokens.

**Compliance With Llm Reviewing Policy:**

Affirmed.

**Final Justification:**

Most of my concerns are addressed, thus I will give a positive score. However, the proposed method is not novel enough in 2026 since there have already emerged a lot of multi-modal latent reasoning papers. Overall, my final decision is borderline accept.

**Key Questions For Authors:**

Is the progressive training/distillation process similar to Coconut?

**Limitations:**

Please refer to Weakness.

**Strengths And Weaknesses:**

Strength: Comprehensive evaluation against multiple baselines and on multiple benchmarks demonstrate the efficiency of Heima, supporting the main contribution.

Weakness:
1. The overall presentation is obsecure, probably due to the complexity of the proposed architecture.
2. There's a lot of missing references/discussion to latent reasoning papers (Coconut, CODI, Colar, SimCoT, etc.)
3. The proposed framework cascading multiple modules, where errors could happen at any stage and accumulate.

---

> ### Author Rebuttal · Authors · 2026-03-30
>
> We thank the reviewer for acknowledging the comprehensive evaluation and the efficiency contribution. We address each concern below.
>
> **W1: Presentation is obscure**
>
> We appreciate this feedback and will improve clarity in the revision. We want to clarify that Heima's architecture is actually simple — the perceived complexity may stem from the presentation of two components (Heima + interpreter) together, when in practice they serve very different roles:
>
> - At inference time, only Heima is used. It is a standard autoregressive MLLM that generates thinking tokens followed by the answer — no architectural modification to the base model, no additional modules, no multi-stage pipeline. It is identical to any standard MLLM, except the model outputs a few special tokens instead of verbose CoT text.
> - The interpreter is used only for post-hoc analysis to validate that thinking tokens encode meaningful reasoning. It is not part of the inference pipeline and does not affect deployment complexity.
>
> In the revision, we will make this more clearly with a concise algorithmic summary of both the training and inference to improve readability.
>
> **W2: Missing references/discussion to latent reasoning papers**
>
> For related works, we provide a brief comparative discussion below and will incorporate a comprehensive discussion in the revision.
>
> Coconut: As discussed in our Section 2.3, Coconut uses continuous hidden states as latent representations on GPT-2 for text-only math tasks. Heima differs by using discrete thinking tokens for large-scale MLLMs, introducing the interpreter for reconstruction-based validation, and providing information-theoretic analysis. While both use progressive training, the loss computation differs — Heima supervises both thinking tokens and remaining textual stages, whereas Coconut masks out latent thoughts.
>
> CODI: CODI proposes a collaborative distillation framework where a large teacher model and a small student model are jointly fine-tuned. It focuses on knowledge distillation between models of different sizes, rather than compressing the reasoning process itself within a single model. Heima addresses a fundamentally different problem — compressing CoT into latent tokens within the same model for inference efficiency.
>
> CoLAR: CoLAR explores latent representations for reasoning in LLMs. However, it does not address multimodal reasoning or provide mechanisms for validating compressed representations. Heima's interpreter framework and information-theoretic grounding are distinct contributions absent from CoLAR.
>
> SimCoT: SimCoT focuses on simplifying chain-of-thought prompting strategies rather than compressing CoT into latent representations. It operates at the prompt/instruction level, whereas Heima operates at the representation level by distilling reasoning stages into thinking tokens.
>
> Key distinction of Heima: Unlike all the above works, Heima uniquely combines: (1) discrete thinking tokens compatible with standard autoregressive inference, (2) application to large-scale MLLMs with multimodal inputs, (3) an interpreter framework that reconstructs and validates compressed reasoning, and (4) information-theoretic analysis of the compression gap. No single prior work addresses all four aspects.
>
> **W3: Cascading modules where errors could accumulate**
>
> We respectfully clarify that Heima does not cascade multiple modules at inference time. The framework has only one model at inference: the Heima MLLM, which generates thinking tokens and the final answer in a single autoregressive pass. There is no multi-stage pipeline where errors propagate between modules.
>
> The interpreter is a separate, offline analysis tool — it is never part of the inference pipeline. Within Heima's single-pass inference, the progressive distillation training does involve stage-by-stage compression, but this is a training strategy, not an inference-time cascade. At inference, all CoT stages are already distilled. There are no intermediate processing or error-prone handoffs between modules.
>
> **Q1: similar to Coconut?**
>
> Both Heima and Coconut adopt progressive training, but the specifics differ significantly:
>
> | Aspect | Coconut | Heima |
> |---|---|---|
> | Representation| Continuous hidden states (bypasses vocabulary) | Discrete thinking tokens (in vocabulary) |
> |Loss computation| Masks out questions and latent thoughts; loss only on remaining textual tokens | Loss on both thinking tokens and remaining textual CoT stages |
> |Recovering stage| Not used | Additional recovering stage after progressive distillation to consolidate cross-stage interactions |
> |Model scale| GPT-2 (124M), text-only | MLLMs (7B-11B), multimodal |
> |Supervision signal| Only future text tokens | Full sequence including thinking tokens |
>
> Progressive training is a general curriculum learning strategy — its use in both methods does not imply methodological equivalence. Our representation design, loss formulation, and recovering stage are all distinct to Heima.

---

> > ### Author Rebuttal · Reviewer_n63E · 2026-04-02
> >
> > Thank you for the retailed response. Most of my concerns are addressed. I decide to raise my score accordingly.

---

### Official Review · Reviewer_brSC · 2026-03-13

**Soundness:** 2
**Presentation:** 3
**Significance:** 2
**Originality:** 2
**Overall Recommendation:** 4
**Confidence:** 3

**Summary:**

This paper proposes a framework for efficient reasoning in large language models by replacing natural-language chain-of-thought (CoT) with a sequence of learned “thinking tokens”. Instead of generating full reasoning steps in text, the model generates special CoT tokens which are then fed back into the model autoregressively. The authors argue that the reasoning information is stored in the hidden states corresponding to these tokens, allowing the model to perform multi-step reasoning while reducing the token cost of explicit chain-of-thought. An additional interpreter model is trained to reconstruct natural language explanations from the hidden representations of the thinking tokens.

**Compliance With Llm Reviewing Policy:**

Affirmed.

**Final Justification:**

After reading the rebuttal and considering the other reviews, I raise my score from weak reject to weak accept. The Qwen generalization results and the clarified distinction from COCONUT address my main novelty concern. The training data alignment argument for single-token optimality is reasonable. My concern about representational ambiguity across problem complexities remains unaddressed empirically. I encourage the authors to provide direct analysis of hidden state diversity in the revision.

**Key Questions For Authors:**

1. Could you clarify why numbers are bolded in Table 1 when they don't represent the highest performance?

2. Given the 3.1% overall accuracy drop and 7.3% drop on MathVista shown in Table 1, did you run any statistical tests to prove this counts as "comparable" performance?

**Strengths And Weaknesses:**

Strenghts:

The goal of reducing the cost of chain-of-thought reasoning is important, and the paper is clearly written. Using an auxiliary LLM interpreter to decode the hidden states back into text is an interesting sanity check.

Weaknesses:

I have several concerns regarding the conceptual novelty, the interpretation of the thinking tokens, and whether the method truly captures reasoning rather than simply providing additional latent compute steps.
1. Novelty and the Discretization Bottleneck: This paper is very similar to the COCONUT paper (Hao et al., 2024), but adds a restrictive constraint on top of the design. By forcing the model to explicitly predict a newly added discrete CoT token for each stage, the architecture forces the hidden state to pass through the LM head and project to a specific token ID. This creates a representational bottleneck compared to purely continuous latent approaches (like COCONUT) that bypass the unembedding matrix entirely to utilize the full continuous space.

2. Potential Ambiguity Across Reasoning Steps: Considering the bottleneck in the design mentioned in the previous point, the framework assigns a shared, unique special token CoT{(k)} across different samples for the k-th stage. This introduces a serious risk of representational ambiguity. Consider step k in two different problems: in a simple example, this step might correspond to the final conclusion, while in a highly complex problem, it might still be in the early middle of the reasoning process. Forcing both scenarios to map to the exact same discrete CoT{(k)} token constrains the model from distinguishing the different functional roles of these steps across varying complexities.

3. Number of reasoning tokens per CoT: The framework appears brittle when adapting to variable reasoning lengths. In Figure 5 (left), the authors present an ablation study demonstrating that using a single thinking token to encode a CoT stage achieves the best performance, and adding more thinking tokens degrades accuracy. Unusually, adding more thinking tokens makes the accuracy drop. Normally, giving an LLM more tokens allows it to scale its test-time compute to solve harder problems.

---

> ### Author Rebuttal · Authors · 2026-03-31
>
> **W1: Novelty**
>
> We respectfully disagree that Heima is simply COCONUT with a restrictive constraint. The two methods differ fundamentally in design philosophy, representation, and scope:
>
> - COCONUT operates in a purely continuous space: it takes the last hidden state directly as the next input embedding, bypassing the vocabulary entirely. It was validated only on GPT-2 (124M) for text-only math tasks.
> - Heima introduces discrete thinking tokens in vocabulary that are explicitly generated during inference. The reasoning information is encoded in the hidden states at these token positions — but the discrete token serves as a structural anchor that enables: standard autoregressive generation without architectural modifications, compatibility with existing inference infrastructure, and clear stage boundaries for interpreter to reconstruct reasoning.
>
> The "bottleneck" is actually a feature, not a bug. The reviewer suggests that projecting through the LM head is restrictive compared to continuous latent states. However, we note:
>
> 1. The reasoning information lives in the hidden states, not in the token ID. The discrete token ID merely identifies which stage of reasoning. The actual reasoning content is carried by the full-dimensional hidden state at that position, which is conditioned on all prior context.
> 2. Empirical validation supports this. Our interpreter successfully reconstructs detailed, input-specific reasoning from hidden states.
> 3. Practical advantages. COCONUT's continuous approach requires modifying the inference pipeline to inject hidden states as embeddings, breaking compatibility with standard autoregressive frameworks. Heima works with unmodified inference code.
>
> Additionally, beyond representation design, Heima contributes the interpreter for validating compressed reasoning and the information-theoretic analysis, which are missing in COCONUT. We also demonstrate effectiveness on large-scale MLLMs (7B–11B) across diverse multimodal benchmarks, a significant step beyond COCONUT's GPT-2 experiments.
>
> **W2: Representational ambiguity**
>
> We appreciate this thoughtful concern, but we believe it rests on a misunderstanding of how thinking tokens function. As discussed above, the actual reasoning content is determined by the hidden state at that token's position, which is uniquely conditioned on each sample's specific input through the attention mechanism. Our interpreters reconstruct sample-specific details (e.g., identifying "BMW" through prue texts), confirming that the hidden states carry distinct per-sample reasoning information.
>
> For the concern about simple vs complex problems mapping to the same stage: the model naturally adapts the information density of each hidden state based on problem complexity, just as a standard LLM produces hidden states of varying information content at any given token position depending on context.
>
> **W3: More thinking tokens**
>
> We agree this deserves explanation. We offer the following analysis:
>
> 1. Training data alignment. Each CoT stage in LLaVA-CoT-100k contains a semantically coherent reasoning unit. When we use a single thinking token, there is a clean one-to-one mapping. When we use multiple tokens per stage, the model must learn to partition a semantic unit across multiple tokens, but there is no natural sub-structure to guide this partitioning.
> 2. Information diffusion. With multiple thinking tokens per stage, the reasoning information is spread across several hidden states. During generation, each subsequent token only attends to previous tokens—so the reasoning information becomes fragmented. A single token concentrates all stage information, providing a cleaner signal.
>
> **Q1: bold number**
>
> We apologize for the confusion. The bolded numbers in Table 1 are intended to highlight Heima's results to distinguish them from baselines, not to indicate the highest performance.
>
> **Q2: "comparable"**
>
> We acknowledge that the term "comparable" should be used more carefully. We clarify our claims:
>
> - Per-dataset analysis: Heima outperforms LLaVA-CoT on MMBench (+2.1%) and achieves near-identical accuracy on AI2D (-0.1%). The larger drops occur on reasoning-intensive benchmarks (MathVista -7.3%, MMVet -6.5%).
> - Efficiency-accuracy trade-off: The 3.1% average accuracy drop comes with a ~94% reduction in generated tokens. We highlight that latency scales linearly with the number of generated tokens and the 94% token reduction translates directly to significant speedup. We believe this trade-off is significant.
> - Compared to no-CoT baseline: Heima achieves +5.9% average accuracy over the base Llama3.2-11B-Vision model while using  much less output tokens, confirming that reasoning is preserved with significantly higher efficiency.
>
> **Limitations**
>
> We will add a limitations section including: Each CoT stage requires a separate interpreter for reconstruction, increasing system complexity, and validation on larger-scale models (70B+) would further establish scalability.

---

> > ### Author Rebuttal · Reviewer_brSC · 2026-04-03
> >
> > Thank you for the detailed reply. The Qwen2.5-VL experiments and the clarification on the novelty distinction from COCONUT are appreciated. The training data alignment explanation for why single tokens outperform multiple tokens is reasonable given LLaVA-CoT-100k's stage structure.
> >
> > My remaining concern is W2: the authors argue that representational ambiguity is resolved through attention-based context conditioning, but this isn't trivial without direct empirical support. Evidence that hidden states at shared token positions are meaningfully distinct across problems of varying complexity would strengthen this claim.
> >
> > Taking into account the overall contribution, scalable efficiency gains on real MLLMs, the interpreter framework, and the generalization shown across architectures, I am willing to raise my score to weak accept. I encourage the authors to include direct analysis of hidden state diversity in the final version.

---

> > > ### Author Response · Authors · 2026-04-03
> > >
> > > We thank the reviewer for the constructive follow-up and the willingness to raise the score. We address the remaining concern on representational ambiguity below.
> > >
> > > We would like to clarify that our dataset directly mitigates the ambiguity concern raised by the reviewer. In our current work, the training dataset (LLaVA-CoT-100k) is well organized with a consistent set of 4 CoT stages across all samples (Summary, Caption, Reasoning, and Conclusion), where each stage serves a fixed and consistent functional role. Therefore, the scenario described by the reviewer does not arise in our current setup.
> > >
> > > Importantly, although each sample shares the same 4-stage structure (Summary, Caption, Reasoning, and Conclusion), the actual detailed reasoning trajectories and steps within the Reasoning stage, may vary across samples in both length and complexity. Our study already demonstrates that using a single thinking token to compress the entire Reasoning stage  (which may internally contain multiple detailed reasoning steps of varying length) is feasible and leads to significant efficiency improvements with competitive performance. This provides empirical evidence that the hidden states at shared token positions are indeed capable of encoding meaningfully distinct reasoning content across problems of varying complexity, even when the underlying reasoning trajectories differ.
> > >
> > > Extending Heima to settings with variable-length CoT stages (i.e., different numbers of stages across samples) would require fine-grained annotation of each sample in the dataset to define sample-specific stage boundaries. We leave this as an important direction for future study.
> > >
> > > Furthermore, we emphasize that Heima is fundamentally a compression framework built on top of distillation. Our method operates as a post-hoc compression layer applied to an already-trained CoT model, without imposing constraints on the underlying CoT structure of the base model. Therefore, if the original uncompressed model can be successfully trained  with CoT stages of various lengths and complexities (which is a property of the base model and its training data, orthogonal to our compression contribution), we believe that our distillation-based compression framework can be  directly applied to such settings as well. The compressibility of each stage depends on the distillation process learning to encode the stage's semantic content into the hidden state — a mechanism that is agnostic to the absolute length or functional role of the original stage, as long as the training data provides consistent stage-level supervision.
> > >
> > > We will include a discussion of this design choice and the future direction of variable-length CoT compression in the revised manuscript, along with the hidden state diversity analysis the reviewer suggested.

---

### Official Review · Reviewer_5wGV · 2026-03-15

**Soundness:** 3
**Presentation:** 3
**Significance:** 3
**Originality:** 3
**Overall Recommendation:** 5
**Confidence:** 4

**Summary:**

Research to shorten CoT in AI. As an idea, we prepared a token dedicated to CoT, and during CoT, we achieved compression by using only that token. In other words, the idea is to tune each token to have the value of a few tokens in the CoT before compression. This is no longer a simulation of human thought processes, but a thought divided into storage-levels for temporarily writing out processes as notes in reasoning.

**Compliance With Llm Reviewing Policy:**

Affirmed.

**Final Justification:**

The rebuttal addresses my concerns and I think all issues is solved.

**Key Questions For Authors:**

-  The size of the Heima model itself could not be read in the text. In other words, if the model size is large enough, even if the CoT is short, the performance should be achieved, and it is not possible to determine whether the CoT is really compressed or not.

- For example, in arithmetic tasks, there is a concern that numerical tokens that are not always included in the training data will appear in the thought process. Although it is not limited to arithmetic,
Of course, it does not appear in the training data, so I think that there is no corresponding CoT token. Is it correct that this language model is missing from CoT?

**Limitations:**

yes

**Strengths And Weaknesses:**

Strengths:
- Soundness: How to compress CoT, and the amount of information that can be compressed, and it is likable by estimating information theoretically.
- Originality: The idea is very interesting. I think that reasoning is not a language, it is very suggestive in the Copernican-like turn.

Weakness:
- (Cont. Soundness) On the other hand, the size of the Heima model itself could not be read in the text. In other words, if the model size is large enough, even if the CoT is short, the performance should be achieved, and it is not possible to determine whether the CoT is really compressed or not.
- Significance: In order to compress information, it is unfortunate that the information to be compressed is required, which means that this compression method has become a domain-specific method. In particular, it is not included in the training data like extrapolation, but it is easy to predict that the performance will fall due to the lack of learning of CoT tokens in a place that should have been reached if you infer carefully.

---

> ### Author Rebuttal · Authors · 2026-03-30
>
> We sincerely thank the reviewer for the positive assessment and the thoughtful comments. We address each concern below.
>
> **W1/Q1: Model size not clearly stated**
>
> We apologize for the lack of clarity. We apply our Heima method to models of different sizes, as shown in Line 308 and 324. We explicitly state the model sizes below:
>
> - Heima (main): Based on LLaVA-CoT / Llama-3.2-11B-Vision-Instruct (11B parameters)
> - Heima (LLaVA family): Based on LLaVA-Next-Vicuna-7B (7B parameters)
> - Interpreters: Llama-3.1-8B-Instruct (8B) or Vicuna-7B (7B), used only for analysis, not during inference
>
> We can also apply our Heima to Qwen family based on Qwen2.5-VL-7B (7B parameters) with results shown in the rebuttal for reviewer nVXS.
>
> Critically, Heima does not increase model size. We apply LoRA adapters (rank 16) during training and use the exact same base model at inference. The only difference between the baseline (LLaVA-CoT) and Heima is the replacement of verbose textual CoTs with thinking tokens — the model architecture and parameter count remain identical.
>
> This directly addresses the concern about whether model capacity rather than compression drives performance. The controlled comparison in Table 1 shows that for the same model (Llama-3.2-11B-Vision), our Heima (compressed CoT) achieves 58.0% average accuracy using only ~6% of tokens on certain tasks, outperforms the non-CoT baseline with 52% average accuracy. Similar observations can also be obtained for LLaVA-Next-Vicuna-7B in table 2. Our method can achieve significantly higher accuracy with few tokens compared with the non-CoT models, demonstrating our effectiveness through compression, rather than simply relying on the inherent capacity of the model to skip reasoning entirely. If the model were just "large enough to not need CoT," it would perform similarly to the no-CoT baseline — but it does not.
>
> **W2: Domain-specific method**
>
> We appreciate this insightful observation. We acknowledge that Heima requires a CoT-annotated training dataset, which ties the compression to domains covered by  the training data. However, we offer the following perspectives:
>
> 1. Broad coverage through diverse training data: The LLaVA-CoT-100k dataset integrates samples from multiple VQA domains (ShareGPT4V, ChartQA, ScienceQA, AI2D,  etc.). Despite training on this single dataset, Heima generalizes well to held-out benchmarks not represented in the training set (e.g., MMVet, HallusionBench, MMStar), achieving competitive accuracy. This suggests the compressed reasoning patterns transfer across related domains rather than being narrowly domain-specific.
>
> 2. Analogous to standard CoT fine-tuning: The requirement for CoT training data is shared by all CoT fine-tuning methods (e.g., LLaVA-CoT, LLaVA-o1). Heima adds compression on top of this existing requirement without introducing additional data needs. Any dataset suitable for CoT fine-tuning can be directly used for Heima training.
>
> 3. Scalability path: As CoT-annotated datasets grow in scale and diversity (a clear trend in the field), Heima's coverage naturally expands. Furthermore, CoT annotations can be generated automatically via API-based distillation from stronger models, making dataset construction increasingly accessible.
>
> We agree this is a meaningful limitation and will discuss it explicitly in the revised limitations section.
>
>
> **Q2: Unseen numerical tokens/out-of-distribution reasoning**
>
> We clarify that thinking tokens do not replace individual numerical or textual tokens. Instead, each thinking token compresses an entire reasoning stage (e.g., the full "Summary" stage or "Reasoning" stage) into a single latent representation.
>
> The thinking tokens (e.g., "Thinking_of_Summary", "Thinking_of_Reasoning") are shared across all samples — the same token is used regardless of the specific content. The reasoning information is encoded not in the token symbol itself, but in the hidden state (last hidden state) produced by the model when it generates that token. This hidden state is conditioned on the specific input (image + question) through the model's attention mechanism, allowing it to capture input-specific reasoning patterns including novel numerical values.
>
> To illustrate: when Heima encounters a math problem with numbers not seen during training, the model processes the image and question through its full attention layers, and the hidden state at the thinking token position encodes the reasoning about those specific numbers — just as a standard LLM can process unseen numbers through its learned arithmetic patterns.
>
> The reviewer's intuition is partially correct: for problems requiring very long or complex reasoning chains that differ substantially from training distribution, performance may degrade. Our results on MathVista (43.6% vs. 50.9% for full CoT) reflect this trade-off, where dense multi-step numerical reasoning is harder to compress. We view this as a meaningful direction for future work.

---

> > ### Author Rebuttal · Reviewer_5wGV · 2026-04-06
> >
> > W1/Q1: Model size not clearly stated
> >
> > I think the problem has been completely solved here. This is due to the oversight of the reviewers. And I also knew that the model size will not be increased from the training method LoRA. This matter has been perfectly resolved.
> >
> > W2: Domain-specific method
> >
> > There was an agreement between the reviewer and the author. This method is domain-specific. The advocating that users can increase the data and take action, as the user shows, leads to an improvement in the score on the benchmark, but it did not overturn the reviewer's judgment in that the improvement of benchmark was not all of the Academia / Industry's research and it was not a definitive solution. However, the sincerity of listing in the recitation is sufficient material to reconsider the improvement of the evaluation point in providing appropriate information to the reader.
> >
> > Q2: Unseen numerical tokens/out-of-distribution reasoning
> >
> > I think it was explained enough. And I was happy that it would be a challenge for the future.

---

### Official Review · Reviewer_nVXS · 2026-03-17

**Soundness:** 1
**Presentation:** 2
**Significance:** 2
**Originality:** 1
**Overall Recommendation:** 2
**Confidence:** 4

**Summary:**

This paper proposes a novel method named Heima, which aims to reduce the number of inference tokens in multimodal tasks by compressing the reasoning process into a latent space.
However, the paper has several major issues: the theoretical analysis is too broad, there is a lack of comparative discussion with related works, the baseline models selected for experiments are weak, and it only reports the reduction in token count while ignoring other important efficiency metrics.

**Compliance With Llm Reviewing Policy:**

Affirmed.

**Key Questions For Authors:**

See weaknesses

**Limitations:**

No discussion on the limitations in the paper. I think it is necessary to add some.

**Strengths And Weaknesses:**

**Strengths**
1.  The paper investigates an interesting question, and the empirical study also demonstrates that inference tokens are significantly reduced.

**Weaknesses**
1.  The theoretical analysis seems unnecessary and appears to explain not only Heima but also the broader scope of reasoning in latent space.
2.  A comparative discussion with related work is lacking, such as with Coconut and Cocomix. Although Heima is applied in a multimodal context, this is not the core innovative point.
3.  The baseline models used are too weak; at the very least, models like Qwen-VL and InternVL should have been included.
4.  In addition to reasoning tokens, other efficiency metrics, such as training cost and inference latency, are not reported.

---

> ### Author Rebuttal · Authors · 2026-03-31
>
> **W1: Theoretical analysis**
>
> We respectfully clarify that our information-theoretic analysis is not intended as a general explanation of latent-space reasoning, but serves as the ante-hoc conceptual foundation that directly motivates and guides our design, as mentioned in Remark 3.2 and Line 254~260. Specifically, Theorem 3.1 formally defines the compression-induced information gap, which serves two critical roles unique to Heima:
>
> 1. **Motivating the compression design**: The theorem establishes that reasoning capability is preserved when non-trivial mutual information is retained, justifying our approach of distilling verbose CoTs into compact thinking tokens.
>
> 2. **Directly guiding the interpreter design**: The interpreter component was explicitly designed to empirically quantify this theoretical gap. By reconstructing textual reasoning from thinking tokens and comparing with original CoTs, we obtain a practical proxy for the information loss defined by the theory. The coupling between theory and architecture is our key contribution.
>
> Our theoretical analysis to formalize CoT compression and to motivate the interpreter as an empirical gap estimator is novel and specific to Heima. Prior latent reasoning works (Coconut or Compressed CoT) lack this theoretical grounding entirely.
>
> **W2: discussion with Coconut and Cocomix**
>
> We appreciate this comment. We show some discussions on Coconut and other works in Section 2. Here we provide a more detailed comparison below and will incorporate the full discussion in the revision.
>
> 1. **Coconut**: Coconut operates on small-scale models (GPT-2) for text-only math tasks, using continuous hidden states (last hidden states without converting to discrete tokens) as latent representations. In contrast, Heima: (1) defines discrete thinking tokens in the vocabulary; (2) targets large-scale MLLMs with multimodal inputs; (3) introduces the interpreter framework for reconstructing and validating compressed reasoning; and (4) provides information-theoretic analysis absent in Coconut. Heima computes loss on both thinking tokens and remaining textual CoT stages, while Coconut masks out questions and latent thoughts.
>
> 2. **Cocomix**: Cocomix explores mixing continuous and discrete representations during language model pre-training, focusing on general language modeling rather than reasoning compression. It does not address CoT compression or multimodal reasoning. Heima is fundamentally different in its goal (efficient reasoning via CoT compression) and design (thinking tokens + interpreter for reconstruction).
>
> 3. **Core distinction**: Our multimodal focus is not merely an "application extension". The interpreter's ability to reconstruct visual information (e.g., identifying "BMW logo" "black exterior") from pure textual inputs demonstrates that thinking tokens encode cross-modal reasoning signals, a capability not explored in any prior latent reasoning work.
>
> **W3: Baseline models are too weak**
>
> LLaVA-CoT is a SOTA reasoning MLLM built on Llama-3.2-11B-Vision. We further conducted experiments with additional Qwen2.5-VL-7B to demonstrate generalization (# denotes average number of tokens):
>
> |Model|MMSar (#)|MMBench (#)|MMVet (#)|MathVista (#)|AI2D (#)|Hallusion (#)|Average|
> |--|--|--|--|--|--|--|--|
> |Qwen2.5-VL-7B|60.3 (52)|80.0 (10)|65.4 (138)|66.7 (203)|80.9 (3)|50.7 (69)|67.3|
> |Qwen2.5-VL-7B CoT|65.2 (182)|82.1 (139)|69.4 (235)|67.8 (204)|84.4 (182)|64.8 (164)|72.3|
> |Heima|61.1 (12)|81.9 (13)|59.5 (72)|58.7 (13)|79.3 (13)|63.1 (16)|67.3|
>
> Qwen results confirm that Heima generalizes beyond Llama-based architectures, achieving accuracy comparable to the non-CoT baseline while using only ~6-8% tokens.
>
> **W4: training cost and inference latency**
>
> **Training Cost**: Heima uses LoRA adapters (rank 16, alpha 32) with frozen visual encoders. LLaVA-CoT-100k has 100K samples with 3 CoT stages, requiring 4 epochs total. On 8x H100 GPUs, this completes in 4-5 hours total. It is a one-time cost that enables persistent inference savings.
>
> **Inference Latency**: For autoregressive generation, latency scales linearly with the number of generated tokens (each token requires one forward pass). Thus, the ~94% token reduction translates directly to significant speedup. We show the latency in the table and observe that token reduction can directly reduce the latency.
> |Number of Generated Tokens|Total Latency (s)|
> |--|--|
> |16|0.2|
> |32|0.4|
> |128|1.6|
> |256|3.3|
> |512|6.6|
>
> **Memory**: Heima uses the same base model architecture with no additional parameters. The reduced sequence length also decreases KV-cache memory usage proportionally.
>
> **W5: limitations**
>
> We will add a limitations section including:
>
> 1. Multiple interpreters: Each CoT stage requires a separate interpreter for reconstruction, increasing system complexity.
> 2. Scale of evaluation: Our experiments are conducted on models up to 11B parameters. Validation on larger-scale models (70B+) would further establish scalability.

---

> > ### Author Rebuttal · Reviewer_nVXS · 2026-04-04
> >
> > Thanks for your response. The current version requires more solid experimental support and a careful consideration of theoretical necessity, so I maintain my score.

---

> > > ### Author Response · Authors · 2026-04-05
> > >
> > > Dear Reviewer nVXS,
> > >
> > >
> > > Thank you for your response. However, we respectfully note that your reply does not specify which aspects of our experimental support remain insufficient or why our theoretical analysis is unnecessary. We would genuinely appreciate concrete feedback so that we can engage in a productive discussion. We address both points below.
> > >
> > >
> > > ## "More solid experimental support"
> > >
> > >
> > > In our rebuttal, we provided substantial new experimental evidence directly responding to your original concerns:
> > >
> > >
> > > 1. **Additional model family (W3):** We presented full benchmark results on **Qwen2.5-VL-7B**, extending our evaluation to three distinct model families (Llama-3.2, Vicuna/LLaVA-Next, Qwen2.5-VL). We also committed to adding InternVL in the revision. Note that validating across **three diverse model families** already provides strong evidence of generalization, which is the underlying concern behind your W3. We would also like to point out that the rebuttal period is limited in duration, and running, validating, and reporting new experiments on additional architectures within this window is a substantial task. We believe the new Qwen2.5-VL results — a model family architecturally distinct from both Llama and Vicuna — meaningfully strengthen the experimental evidence.
> > > 2. **Training cost (W4):** We reported that Heima was trained in 4–5 hours on 8×H100 GPUs using lightweight LoRA adapters — a one-time cost enabling persistent inference savings.
> > > 3. **Inference latency (W4):** We provided wall-clock latency measurements showing that the ~94% token reduction translates directly to proportional speedup, with a latency table included.
> > > 4. **Memory analysis (W4):** We clarified that Heima adds no parameters at inference and reduces KV-cache memory proportionally to the token reduction.
> > >
> > > Could you please clarify which of these responses you find insufficient, and what specific additional experiments or metrics would change your assessment? Without this specificity, it is difficult for us to understand what "more solid experimental support" means in practice, given that we have directly addressed every experimental concern in your original review.
> > >
> > >
> > > ## "Careful consideration of theoretical necessity"
> > >
> > >
> > > In our rebuttal, we explained in detail that our information-theoretic analysis is not a post-hoc justification but the **ante-hoc conceptual foundation** that directly motivated and guided two key design decisions in Heima:
> > >
> > >
> > > 1. **Theorem 3.1** formalizes the compression-induced information gap, which **motivates** the thinking token design by establishing when reasoning capability is preserved under compression.
> > > 2. The same theorem **directly guides the interpreter design** — the interpreter was explicitly built to empirically estimate this theoretical gap by reconstructing textual reasoning from thinking tokens.
> > >
> > >
> > > Our information-theoretic analysis is not meant to be a general, standalone analysis of all latent-space reasoning; rather, it serves strictly as the **motivation and conceptual foundation** that directly drives our specific architectural and interpreter design.
> > > This tight coupling between theory and architecture is a contribution absent from all prior latent reasoning works (Coconut, Compressed CoT, etc.). We note that Reviewer 5wGV explicitly found the theoretical analysis "sound" and described our approach as a "Copernican-like turn."
> > >
> > >
> > > Could you please specify what you mean by "theoretical necessity"? Do you believe the theorems contain errors? That the connection between theory and method design is unclear? Or that the paper would be stronger without the theoretical section entirely? Understanding your specific concern would allow us to respond constructively.
> > >
> > >
> > > ## Summary
> > >
> > >
> > > We have made every effort to engage thoroughly with your feedback, providing new experiments, efficiency metrics, and a limitations discussion — all directly addressing your five listed weaknesses. We respectfully ask that you engage with the specific content of our rebuttal and clarify which responses you find inadequate and why, so that we may continue a productive dialogue. A blanket statement to "maintain my score" without addressing any of our detailed responses does not give us an opportunity to resolve your concerns.
> > >
> > > We remain fully open to further discussion and any additional clarifications you may need.

---

### Decision · Program_Chairs · 2026-04-30

**Decision:**

Accept (regular)

**Comment:**

This paper proposes a method for compressing CoT into a small number of discrete "thinking tokens", which enables efficient implicit CoT reasoning in multimodal LLMs. The approach achieves reductions in token usage while maintaining competitive accuracy across multiple benchmarks and model families. Reviewers generally agree that the problem is important and the empirical results are strong. In particular, the method is validated on modern multimodal LLMs (7B-11B scale), and the efficiency-accuracy tradeoff is compelling. The use of discrete thinking tokens, which preserves compatibility with standard autoregressive inference pipelines, provides a practical instantiation of implicit CoT, in contrast to prior continuous-state approaches.

While there are concerns regarding the level of novelty relative to recent implicit CoT works and the necessity of the theoretical analysis, the rebuttal clarifies these and strengthens the empirical evidence by adding experiments on Qwen-based models. The remaining limitations (e.g., potential representational bottlenecks and performance drops on more complex tasks) are acknowledged.

Overall, this paper presents an interesting approach to efficient implicit CoT reasoning in multimodal models, and reviewers mostly lean towards acceptance (1 accept, 2 weak accepts, and 1 reject), therefore I recommend weak acceptance.